

# Twistor space origins of the Newman-Penrose map

**Kara Farnsworth[1]\*, Michael L. Graesser[2]† and Gabriel Herczeg[3]‡**

**1** CERCA, Department of Physics, Case Western Reserve University,
Cleveland, OH 44106, USA
**2** Theoretical Division, Los Alamos National Laboratory, Los Alamos, NM 87545, USA
**3** Brown Theoretical Physics Center, Department of Physics, Brown University,
Providence, RI 02912, USA

\* kmfarnsworth@gmail.com, † michaelgraesser@gmail.com, ‡ gabriel_herczeg@brown.edu

## Abstract

Recently, we introduced the "Newman-Penrose map", a novel correspondence between a certain class of solutions of Einstein's equations and self-dual solutions of the vacuum Maxwell equations, which we showed was closely related to the classical double copy. Here, we give an alternative definition of this correspondence in terms of quantities that are defined naturally on twistor space, and a shear-free null geodesic congruence on Minkowski space whose twistorial character is articulated by the Kerr theorem. The advantage of this reformulation is that it is purely geometrical in nature, being manifestly invariant under both spacetime diffeomorphisms and projective transformations on twistor space. While the original formulation of the map may be more convenient for most explicit calculations, the twistorial formulation we present here may be of greater theoretical utility.

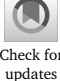
# 1 Introduction

Some of our deepest insights into gauge theories and gravity have emerged from the study of correspondences relating the two. One such promising relationship is the double copy, which can be summarized schematically by the equation

$$\text{gravity} = \text{gauge} \otimes \text{gauge}. \tag{1}$$

This general structure has been revealed in many contexts over the past few decades. The prehistory of the double copy began in 1986, when Kawai, Lewellen and Tye [1] realized that there was a close relationship between the tree-level amplitude of a closed string and the square of the tree-level amplitude of an open string. The subject began in earnest a little more than twenty years later, when Bern, Carrasco and Johansson [2] observed that a similar relationship existed between perturbative amplitudes in gauge theories and gravity. Their results have now been proven at tree-level [3], and a growing body of evidence suggests that they can be extended beyond tree-level in perturbation theory [4–11]. With such promising evidence at the amplitude level, it is natural to look for a classical formulation of the double copy in order to understand how fundamental this relationship between gauge theory and gravity is.

Although there have been attempts at a Lagrangian formulation [12–14], much recent success has been found realizing double copies in a classical setting, as maps between exact solutions of gauge theories and gravity. Several such maps have been proposed and explored [15–40], and many of them share the common feature that null geodesic congruences play a prominent role. For example, in the Kerr-Schild double copy of [15], the gauge field is itself the dual of the tangent vector to a geodesic null congruence. The Newman-Penrose map introduced in [16] also has this flavor, where the gravity solutions one considers are of Kerr-Schild type, and the Kerr-Schild vector is assumed to be tangent to a shear-free null geodesic congruence (SNGC) with non-vanishing expansion. In the Weyl double copy of [17], the existence of an SNGC is guaranteed by virtue of the Goldberg-Sachs theorem for algebraically special spacetimes [41]. In light of the Kerr Theorem [42–45], which realizes any SNGC in Minkowski space as the solution of an equation defined by the vanishing of a holomorphic, homogeneous function of twistor variables, it is perhaps not surprising that the Weyl double copy was recently given an elegant twistorial formulation [46,47] that can also accommodate algebraically special *linearized* solutions of *any* Petrov type.

One might wonder whether the Newman-Penrose map can also be given a twistorial formulation. The primary purpose of this article is to answer this question in the affirmative: We provide a twistorial formulation of the Newman-Penrose map that is manifestly invariant under both spacetime diffeomorphisms and projective transformations on twistor space.

This paper is organized as follows: In Section 2 we review the Newman-Penrose map, originally defined in [16]. In Section 3 we give an overview of the relevant aspects of the two-component spinor and twistor formalisms, and in Section 4 we present the twistor space version of the Newman-Penrose map. Finally, we conclude in Section 5 and discuss the consequences of the choice of SNGC normalization in Appendix A.

# 2 The Newman-Penrose map

## 2.1 Kerr-Schild spacetimes

Here we review the salient features of the Newman-Penrose map as defined in [16]. We begin by recalling some details about Kerr-Schild spacetimes and their construction in the Newman-

Penrose formalism. A Kerr-Schild solution of Einstein's equations is one that can be written in the form

$$g_{\mu\nu} = \eta_{\mu\nu} + V l_\mu l_\nu, \tag{2}$$

where $\eta_{\mu\nu}$ is a flat metric, $l_\mu$ is a null vector,[1] and $V$ is a scalar function. If we assume that $l^\mu$ is geodesic, shear-free, and expanding (see section 3.2 of [16] for a concise discussion of shear and expansion) then we can write simple expressions defining a null tetrad for the metric (2) [48, 49]:

$$
\begin{aligned}
l &= \partial_v - \Phi \partial_\zeta - \bar\Phi \partial_{\bar\zeta} + \Phi\bar\Phi \partial_u, \\
n &= \partial_u - \tfrac{1}{2} V l, \\
m &= \partial_{\bar\zeta} - \Phi \partial_u, \\
\bar m &= \partial_\zeta - \bar\Phi \partial_u,
\end{aligned}
\tag{3}
$$

where $u, v$ are real light-cone coordinates, and $\zeta, \bar\zeta$ are complex conjugate coordinates related to the usual Cartesian coordinates $(t, x, y, z)$ by

$$u = \tfrac{1}{\sqrt 2}(t - z), \quad v = \tfrac{1}{\sqrt 2}(t + z), \quad \zeta = \tfrac{1}{\sqrt 2}(x + iy), \tag{4}$$

and $\Phi(u, v, \zeta, \bar\zeta)$ is a complex scalar. In these coordinates, the flat metric takes the form

$$\eta_{\mu\nu} dx^\mu dx^\nu = 2(du\,dv - d\zeta\,d\bar\zeta). \tag{5}$$

In terms of the null tetrad (3), we can write the inverse metric as

$$g^{\mu\nu} = l^\mu n^\nu + n^\mu l^\nu - m^\mu \bar m^\nu - \bar m^\mu m^\nu = \eta^{\mu\nu} - V l^\mu l^\nu. \tag{6}$$

The shear-free and geodesic conditions on $l^\mu$ are equivalent to the following nonlinear partial differential equations for $\Phi$:

$$\Phi_{,v} = \Phi\Phi_{,\zeta}, \qquad \Phi_{,\bar\zeta} = \Phi\Phi_{,u}. \tag{7}$$

The equations (7) together imply that the complex scalar $\Phi$ is harmonic with respect to the flat metric $\eta_{\mu\nu}$, so that

$$\Box_0 \Phi = 2(\Phi_{,uv} - \Phi_{,\zeta\bar\zeta}) = 0. \tag{8}$$

When the spacetime solves the vacuum Einstein equations, the function $V$ can be solved for in terms of $\Phi$, so that any $\Phi$ satisfying (7) completely specifies the solution up to constants of integration.

## 2.2 Newman-Penrose map

The spin-raising operator

$$\hat k = -\frac{Q}{2\pi\epsilon_0}(dv\,\partial_\zeta + d\bar\zeta\,\partial_u) \tag{9}$$

was used in [15, 22] to define the self-dual double copy. Here we have included the constants $Q$, representing the charge, and $\epsilon_0$, representing the vacuum permittivity, to directly map to solutions of Maxwell's equations. In that work, the authors justified the form of the operator by observing that it satisfied certain properties that made it analogous to a "momentum-space" version of the Kerr-Schild vector $l^\mu$. However, this is not the unique operator possessing those

---

[1]Note that if $l_\mu$ is null with respect to either $g_{\mu\nu}$ or $\eta_{\mu\nu}$ then it is null with respect to both.

properties, and the particular choice (9) is not justified over any other similar choice of operator. In fact, other possible choices of operator can easily be found, e.g., by substituting either $u \leftrightarrow v$, or $\zeta \leftrightarrow \bar{\zeta}$, or both, in (9).[2]

It is not a new observation that spin-raising operators such as the one appearing in (9) map solutions of the wave equation (often called 'Hertz'-type potentials in this context) to solutions of Maxwell's equations. In addition to their appearance in the self-dual double copy [15, 22], spin-raising operators of a very similar form have also appeared in [50], in which scattering of plane wave solutions was studied in the context of the double copy. Much earlier, a rather general spinorial version of the spin-raising operators that takes a solution of the zero rest-mass equations with spin $s$ to a solution of the zero rest-mass equations with spin $s + 1/2$ was described by Penrose in [51] (see also [52]); in fact, this construction was used to give a spinorial description of the Newman-Penrose map in appendix B of [16]. It has also been known for some time that spin-raising and spin-lowering operators that map solutions of the zero rest-mass equations to solutions with arbitrarily raised or lowered spins can be associated with a choice of twistor of an appropriate rank and type (see section 6.4 of [53], for example). While all of these structures are related to the Newman-Penrose map, we emphasize that the identification of a complex solution of the wave equation associated with a given *real* solution of the Einstein equations of Kerr-Schild type, upon which such a spin-raising operator may act to generate a solution of the vacuum Maxwell equations, is a novel feature of the Newman-Penrose map. The connection with twistor theory that we present here is similarly novel: while the twistorial spin-raising operators of [53] depend on an arbitrary choice of twistor, the construction we present here is defined for a subset of twistors, and is *independent* of the choice of twistor within that subset.

In [16] we used this operator to define the Newman-Penrose map as follows: Given a Kerr-Schild spacetime with an expanding, shear-free Kerr-Schild vector, we can fix the null tetrad in the form (3), and read off the complex function $\Phi$, which satisfies the non-linear partial differential equations (7) and consequently is harmonic. Then the gauge field defined by

$$A = \hat{k}\Phi \tag{10}$$

is necessarily a self-dual solution of the vacuum Maxwell equations, since we have

$$A = -\frac{Q}{2\pi\epsilon_0}\left(\Phi_{,\zeta}dv + \Phi_{,u}d\bar{\zeta}\right), \tag{11}$$

from which we can compute the field strength two-form $F = dA$:

$$F = -\frac{Q}{2\pi\epsilon_0}\left(\Phi_{,u\zeta}du \wedge dv - \Phi_{,\zeta\zeta}dv \wedge d\zeta + \Phi_{,uu}du \wedge d\bar{\zeta} + \Phi_{,u\zeta}d\zeta \wedge d\bar{\zeta}\right), \tag{12}$$

where we used $\Phi_{,uv} - \Phi_{,\zeta\bar{\zeta}} = \frac{1}{2}\Box_0\Phi = 0$. The field strength in (12) is self-dual with respect to $\star_0$, the Hodge star operator associated with the background metric $\eta_{\mu\nu}$—that is, $F$ satisfies

$$F = i \star_0 F, \tag{13}$$

which, together with the fact that $F$ is exact, implies the vacuum flat-space Maxwell equation

$$d \star_0 F = 0. \tag{14}$$

The primary purpose of this article is to give a purely geometric interpretation of the origin of the spin-raising operator (9) and $\Phi$ in terms of structures naturally defined on projective twistor

---

[2]In [40], it was pointed out that this operator, which they used to study the symmetries of the self-dual sectors of Yang-Mills theory and gravity, is closely related to the area form on certain two-dimensional (complex) subspaces of (complex) Minkowski space.

space. This gives a geometric foundation to the coordinate-dependent description which we have reviewed in this section, and we hope that it will provide some additional insight on the self-dual double copy of [15, 22], and the recent discovery of the twistorial origins of the Weyl double copy based on the Penrose transform [46, 47].

# 3 Aspects of twistor theory

Twistors are natural objects for studying null geodesics in complexified Minkowski space $\mathbb{CM}$. Here we briefly review the relevant aspects of twistor theory, focusing on the relationships between geometric structures on $\mathbb{CM}$, and the corresponding geometric strucures on *twistor space* $\mathbb{T}$. Our discussion follows closely the approach taken in chapter 7 of [54]. See also [53, 55, 56] for more background and [57] for a more recent review of twistor theory.

## 3.1 Spin space

In four spacetime dimensions, Minkowski space—or more generally, the tangent space $T_p\mathbb{N}$ to any point $p$ of a pseudo-Riemannian manifold $\mathbb{N}$ of signature $(3, 1)$—is isomorphic to the set of $2 \times 2$ Hermitian matrices. One can make this isomorphism explicit by specifying a soldering form[3] $\boldsymbol{\sigma}_\mu$ with components $\sigma_\mu^{AA'}$, which assigns a Hermitian matrix $V = \boldsymbol{\sigma}_\mu V^\mu$ with components $V^{AA'} = \sigma_\mu^{AA'} V^\mu$ to every real spacetime vector $V^\mu$. Using standard Cartesian coordinates on Minkowski space, a common choice is $\boldsymbol{\sigma}_\mu = (\mathbf{I}, \boldsymbol{\sigma}_i)$, where $\boldsymbol{\sigma}_i$ are the Pauli matrices normalized so that $\{\boldsymbol{\sigma}_i, \boldsymbol{\sigma}_j\} = \delta_{ij}\mathbf{I}$. Such a soldering form maps an arbitrary vector $V^\mu = (V^0, V^i)$ to the Hermitian matrix

$$V = \frac{1}{\sqrt{2}} \begin{pmatrix} V^0 + V^3 & V^1 + iV^2 \\ V^1 - iV^2 & V^0 - V^3 \end{pmatrix}, \tag{15}$$

with $\det(\mathbf{V}) = \frac{1}{2}|V|^2$. Then for any $\mathbf{U} \in SL(2, \mathbb{C})$, the Hermitian matrix $\mathbf{V}' = \mathbf{U}^\dagger \mathbf{V} \mathbf{U}$, has determinant $\det(\mathbf{V}') = \det(\mathbf{V}) = \frac{1}{2}|V|^2$, meaning $\mathbf{U}$ corresponds to a linear transformation preserving the norm of vectors on Minkowski space, i.e. a Lorentz transformation.

We define *spin space*, $\mathbb{S}$, as the space of complex-valued, two-component column vectors on which the elements $\mathbf{U}$ act by left multiplication. A vector $\alpha \in \mathbb{S}$ is called a *spinor*, and it's components are denoted by $\alpha^A$. Similarly, the complex conjugate space is denoted by $\bar{\mathbb{S}}$, and the components of a conjugate spinor are written with primed indices, e.g. $\beta^{A'}$.

Given any spinor $\alpha^A$, one can consider the Hermitian matrix with components $\alpha^A \bar{\alpha}^{A'}$, which has vanishing determinant, so that $\alpha^A \bar{\alpha}^{A'}$ corresponds to a real, null vector. Thus, a spinor can be viewed as the "square root of a null vector." More generally, given $\lambda^A \in \mathbb{S}$, $\mu^{A'} \in \bar{\mathbb{S}}$, the non-Hermitian matrix $\lambda^A \mu^{A'}$ corresponds to a complex null vector. Spinor indices are raised and lowered using the antisymmetric, rank two spinor tensor $\epsilon_{AB}$ via

$$\alpha_B = \alpha^A \epsilon_{AB}, \qquad \alpha^A = \epsilon^{AB} \alpha_B, \tag{16}$$

and every spinor satisfies $\alpha_A \alpha^A = 0$. It is often convenient to fix a *normalized dyad* for spin space $\mathbb{S}$, i.e. a pair of spinors $(o^A, \iota^A)$ satisfying $o_A \iota^A = 1$. A normalized dyad on $\mathbb{S}$ naturally defines a null tetrad on $\mathbb{M}$ via

$$l_0^{AA'} = o^A \bar{o}^{A'}, \quad n_0^{AA'} = \iota^A \bar{\iota}^{A'}, \quad m_0^{AA'} = o^A \bar{\iota}^{A'}, \tag{17}$$

where we have decorated the tetrad elements with subscripts to emphasize that this is a tetrad for flat Minkowski space $\mathbb{M}$, in contrast to the tetrad (3) for the full, Kerr-Schild spacetime.

---

[3]For the considerations of the present article, it will suffice to consider a particular soldering form that is adapted to Minkowski space. Solderings for a general spacetime can be specified by contracting the flat-space soldering form with an orthonormal frame for the curved spacetime metric, but we will not need such structures here.

## 3.2 Twistor space

So far, the discussion has focused only on spinors at a point, but all of these structures can be generalized to smoothly varying functions of $\mathbb{CM}$, or more general four-dimensional manifolds. In particular, we may consider *spinor fields* $\alpha^A(x)$ over $\mathbb{CM}$. The covariant derivative $\nabla_\mu$ on $\mathbb{CM}$ can be extended to a covariant derivative on spinor fields satisfying $\nabla_{AA'}\epsilon_{BC} = 0 = \nabla_{AA'}\epsilon_{B'C'}$.

A *twistor* is a spinor field $\Omega^A(x)$ satisfying the twistor equation

$$\nabla_{A'}{}^{(A}\Omega^{B)} = 0\,. \tag{18}$$

Over complexified Minkowski space $\mathbb{CM}$, (18) can be solved exactly to give

$$\Omega^A = \omega^A - ix^{AA'}\pi_{A'}\,, \tag{19}$$

where $\omega^A$ and $\pi_{A'}$ are constant spinors. Thus, the space of solutions to the twistor equation—i.e., the twistor space, $\mathbb{T}$—is coordinatized by a pair of spinors $Z = (\omega^A, \pi_{A'})$ and we can regard $\mathbb{T}$ as a four-dimensional complex vector space. Twistor space can be equipped with a natural Hermitian inner product, and in particular, each twistor $Z \in \mathbb{T}$ can be assigned a norm

$$|Z|^2 := \omega^A\bar{\pi}_A + \bar{\omega}^{A'}\pi_{A'}\,. \tag{20}$$

The (squared) norm of a twistor can be positive, negative or zero, so we obtain a decomposition of the twistor space $\mathbb{T} = \mathbb{T}_+ \cup \mathbb{T}_0 \cup \mathbb{T}_-$, where $\mathbb{T}_+$ is the set of twistors with positive norm, $\mathbb{T}_0$ is the set of twistors with zero norm, which are called *null twistors*, and $\mathbb{T}_-$ is the set of twistors with negative norm. In what follows, we will be particularly interested in null twistors, which we will see determine real, null geodesics in $\mathbb{CM}$.

In order to understand the relationship between null twistors and real, null geodesics, consider the subset of $\mathbb{CM}$ defined by the zero locus of an arbitrary twistor

$$\Omega^A(x) = 0\,. \tag{21}$$

The general solution (19) then specifies this subset as the points $x^{AA'} \in \mathbb{CM}$ that satisfy the incidence relation

$$\omega^A = ix^{AA'}\pi_{A'}\,. \tag{22}$$

Given a particular solution $x_0^{AA'}$ to (22), the general solution can be written as

$$x^{AA'} = x_0^{AA'} + \lambda^A\pi^{A'}\,, \tag{23}$$

where $\lambda^A$ is an arbitrary spinor. Equation (23) describes a totally null two-plane in $\mathbb{CM}$, called an *$\alpha$-plane*. Every tangent vector to an $\alpha$-plane is null and orthogonal to every other tangent vector. Such a plane cannot exist in real Minkowski space, so the points contained in the plane will generically be complex.

Since (23) is the general solution to (22), the twistor $tZ = (t\omega^A, t\pi_{A'})$ defines the same $\alpha$-plane as $Z$ for any $t \in \mathbb{C}$. The equivalence relation $\sim$ on $\mathbb{T}$, defined by $tZ \sim Z$ for some $t \in \mathbb{C}$, gives rise to the notion of *projective twistor space* $\mathbb{PT} := \mathbb{T}/\sim$. We may then regard the points of $\mathbb{PT}$ as being labeled by $\alpha$-planes.[4] The intersection of the $\alpha$-plane of a null twistor $Z \in \mathbb{T}_0$ and the real subset $\mathbb{M}$ of $\mathbb{CM}$ is described by a real, null geodesic, as we illustrate in Fig. 1 and now show.

We have already observed that the points on an $\alpha$-plane are generically complex. Suppose that some twistor $Z$ defines an $\alpha$-plane that contains at least one real point. Then, without loss of generality, let us assume $x_0^{AA'}$ is real. Contracting (22) with $\bar{\pi}_A$ then gives

$$\omega^A\bar{\pi}_A = ix_0^{AA'}\bar{\pi}_A\pi_{A'}\,. \tag{24}$$

---

[4]$\mathbb{PT}$ is a 3-dimensional complex manifold and can be realized as an open subset of $\mathbb{CP}^3$. The projective space $\mathbb{PT}_0$ of null twistors on $\mathbb{PT}$ is a five-dimensional real manifold.



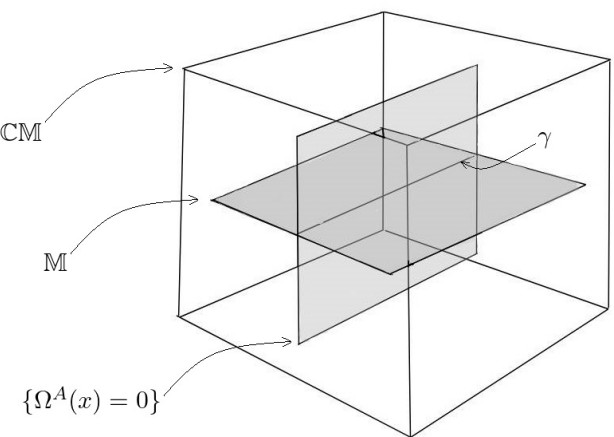

Figure 1: $\mathbb{M}$ and an $\alpha$-plane $\{\Omega^A(x) = 0\}$ as subsets of $\mathbb{CM}$. Their intersection is a real light ray $\gamma$.

Since $x_0^{AA'}$ is real, the right hand side of (24) is clearly imaginary:

$$\omega^A \bar{\pi}_A = ia, \qquad a \in \mathbb{R}. \tag{25}$$

Then we immediately find that

$$|Z|^2 = \omega^A \bar{\pi}_A + \bar{\omega}^{A'} \pi_{A'} = 0. \tag{26}$$

Thus, whenever the $\alpha$-plane defined by a twistor $Z$ contains a real point, $Z$ is null. The converse can be proven as well (see chapter 7 of [54] for details) so $Z$ is null if and only if its $\alpha$-plane contains at least one real point. Moreover, if $x_0^{AA'}$ is real, then the $\alpha$-plane contains the whole real null geodesic

$$x^{AA'} = x_0^{AA'} + r\bar{\pi}^A \pi^{A'}, \quad r \in \mathbb{R}, \tag{27}$$

which establishes that null twistors determine real null geodesics in $\mathbb{CM}$, as claimed. In the following section we introduce a twistorial definition of the Newman-Penrose map that is projectively invariant, and so may be defined directly on $\mathbb{PT}$.

## 4 Newman-Penrose map from twistor space geometry

Consider a null twistor $Z = (\omega^A, \pi_{A'})$ with the property that

$$l_{AA'} \bar{\pi}^A \pi^{A'} = 1, \tag{28}$$

where $l_{AA'}$ is the expanding, null, shear-free and geodesic Kerr-Schild vector. Note that it is always possible to choose a scale for $l_{AA'}$ so that such a twistor exists, and we suppose that such a scale has been chosen. In fact, as we shall see explicitly below, this condition defines a 4-dimensional real subspace of $\mathbb{T}_0$. Moreover the condition (28) has a clear geometric interpretation. The vector $\bar{\pi}^A \pi^{A'}$ is a tangent vector to the real null geodesic contained in the $\alpha$-plane corresponding to $Z$. This can be extended to all of spacetime via parallel transport with respect to the flat metric $\eta_{\mu\nu}$. The normalization condition then requires that this tangent vector be everywhere transverse to $l_{AA'}$ and scaled so that their inner product is normalized to

unity. Kerr's Theorem [42–45] implies that, without loss of generality, we can write[5]

$$l_{AA'}dx^{AA'} = du + \bar{\Phi}d\zeta + \Phi d\bar{\zeta} + \Phi\bar{\Phi}dv, \tag{29}$$

in which case the normalization condition (28) implies that $\bar{\pi}^A\pi^{A'}\partial_{AA'} = \partial_u$. It is also possible to choose the normalization so that the coefficient of $dv$ is equal to one in (29), however we show in the Appendix that this choice leads to the same field strength we obtain using the current normalization.

In order to write explicit expressions in terms of $\omega^A$, and $\pi_{A'}$, we introduce a constant, normalized dyad $o_A\iota^A = 1$ with

$$o^A\bar{o}^{A'}\partial_{AA'} = \partial_v, \quad o^A\bar{\iota}^{A'}\partial_{AA'} = \partial_\zeta, \quad \iota^A\bar{\iota}^{A'}\partial_{AA'} = \partial_u. \tag{30}$$

Note that the dyad $(o^A, \iota^A)$ is tied to a tetrad for flat Minkowski space, in contrast to the tetrad for the full Kerr-Schild spacetime (3). In this basis, the null Kerr-Schild vector (29) can be written as $l_{AA'} = \tilde{o}_A\bar{\tilde{o}}_{A'} = (o_A - \bar{\Phi}\iota_A)(\bar{o}_{A'} - \Phi\bar{\iota}_{A'})$.

Now we can expand the constant spinors $\omega^A$, and $\pi_{A'}$ as

$$\omega^A = \alpha o^A + \beta\iota^A, \qquad \pi_{A'} = \gamma\bar{o}_{A'} + \delta\bar{\iota}_{A'}, \tag{31}$$

where $\alpha, \beta, \gamma, \delta \in \mathbb{C}$.[6] The normalization condition (28) then tells us that

$$\gamma = 0, \qquad |\delta|^2 = 1. \tag{32}$$

Moreover, since $Z$ is null, we have

$$\omega^A\bar{\pi}_A = ia, \quad a \in \mathbb{R}, \tag{33}$$

which implies that $\alpha = -ia\delta$. Now our expessions for the spinor components of $Z$ become

$$\omega^A = -i\delta a o^A + \beta\iota^A, \qquad \pi_{A'} = \delta\bar{\iota}_{A'}. \tag{34}$$

In the generic case when $a \neq 0$, the $\alpha$-plane associated with $Z$ can be described by the equation

$$x^{AA'} = \frac{1}{a}\omega^A\bar{\omega}^{A'} + \lambda^A\pi^{A'}, \tag{35}$$

for arbitrary $\lambda^A$. The special case $a = 0$ requires some care, as it includes both ordinary $\alpha$-planes and $\alpha$-planes that contain real geodesics that are generators of null infinity rather than geodesics on the interior of Minkowski space. Given these subtleties, we will only consider twistors for which $a \neq 0$ from now on. For reasons that will soon be clear, we expand $\lambda^A$ in terms of the spinor dyad as

$$\lambda^A = \bar{\delta}\left(\zeta o^A + \left(u + u_0 - \frac{|\beta|^2}{a}\right)\iota^A\right), \tag{36}$$

where $u_0$ is a real constant, and $u$ and $\zeta$ vary over all complex values. Using this expansion for $\lambda^A$, the equation for the $\alpha$-plane becomes

$$x^{AA'} = v_0 o^A\bar{o}^{A'} + (\zeta + \zeta_0)o^A\bar{\iota}^{A'} + \bar{\zeta}_0\iota^A\bar{o}^{A'} + (u + u_0)\iota^A\bar{\iota}^{A'}, \tag{37}$$

where we have identified $a = v_0$, $\beta = -i\delta\bar{\zeta}_0$. This is then the $\alpha$-plane defined by $v = v_0$, $\bar{\zeta} = \bar{\zeta}_0$, and the tangent bivector $\tau$ to any such plane satisfies the proportionality

$$\tau \propto \partial_u \wedge \partial_\zeta. \tag{38}$$

---

[5]This corresponds to a shift in the dyad (17) of $o_A \to \tilde{o}_A = o_A - \bar{\Phi}\iota_A$, as explained in [58] and appendix A.2 of [16].

[6]These parameters are not the spin coefficients of the tetrad.

Lowering an index with the flat metric leads to

$$\tau^{\mu\rho}\eta_{\rho\nu}dx^{\nu}\partial_{\mu} \propto -(d\bar{\zeta}\partial_{u} + dv\partial_{\zeta}) \propto \hat{k}\,. \tag{39}$$

The above discussion shows that we are able to motivate the form of the operator $\hat{k}$ geometrically from twistor space. However, we would like to describe *all* quantities in the NP map in terms of objects naturally associated to the null twistor $Z$ and the dual to the Kerr-Schild SNGC $l_{AA'}$ without reference to any particular coordinate system. For this, we can introduce the vectors

$$p^{AA'} = \bar{\pi}^{A}\pi^{A'}\,, \quad q^{AA'} = \omega^{A}\pi^{A'}\,, \tag{40}$$

which span the tangent space to the $\alpha$-plane. Now we can take

$$\tau = \frac{1}{\omega^{A}\bar{\pi}_{A}}p \wedge q = -\delta^{2}\partial_{u} \wedge \partial_{\zeta}\,, \tag{41}$$

and define

$$\hat{\kappa} := -\frac{Q}{2\pi\epsilon_{0}}\tau^{\mu\rho}\eta_{\rho\nu}dx^{\nu}\partial_{\mu} = -\frac{Q}{2\pi\epsilon_{0}}\delta^{2}(dv\,\partial_{\zeta} + d\bar{\zeta}\,\partial_{u})\,, \tag{42}$$

which matches the operator (9) up to an overall, arbitrary phase $\delta^{2}$. Next, we need an invariant definition of $\Phi$. For this, we note that

$$\Psi := \frac{1}{\omega^{A}\bar{\pi}_{A}}l_{BB'}\bar{q}^{BB'} = \bar{\delta}^{2}\Phi + \frac{\bar{\delta}\bar{\beta}}{ia}\,. \tag{43}$$

The first term in $\Psi$ differs from $\Phi$ by the constant phase $\bar{\delta}^{2}$, complementary to the phase of $\hat{\kappa}$. The second term is a constant, so it is annihilated by $\hat{\kappa}$. Hence, we can define the NP map in a manifestly invariant manner, depending only on the SNGC $l_{AA'}$ and the twistor $Z$ by

$$A = \hat{\kappa}\Psi\,. \tag{44}$$

In particular, note that this definition of $A$ is independent of the phase $\delta$, the individual dependence in $\Psi$ and $\hat{\kappa}$ having canceled, and is equivalent to the original definition in (10). In fact, if we insist that the normalization condition (28) be preserved under rescalings $Z \to tZ$, $t \in \mathbb{C}$, so that $l_{AA'} \to |t|^{-2}l_{AA'}$, then it can be shown that the Newman-Penrose map depends only on the equivalence class $[Z] = [tZ], t \in \mathbb{C}$, and so is projectively invariant.

We end with a comment about the uniqueness of $\hat{\kappa}$. In general one may consider other operators similar to $\hat{\kappa}$, as described in Section 2.2, and it isn't clear why this particular form of $\hat{\kappa}$ is preferred. The twistor space construction of the NP map described here offers some answers. At the outset, a normalization for $l$ is made to set one of its coefficients equal to a constant, but which coefficient to choose is seemingly arbitrary. To keep $l_{AA'}$ real, the only choices are to set the coefficient of $dv$ or $du$ equal to a constant. Choosing $du$ leads to the operator $\hat{\kappa}$, through the tangent bivector of the $\alpha$-plane associated to projective twistor $[Z]$ described above. If we instead choose $dv$, the tangent bivector is proportional to $\partial_{\nu} \wedge \partial_{\bar{\zeta}}$, leading to a new differential operator proportional to $du\,\partial_{\bar{\zeta}} + d\zeta\,\partial_{\nu}$. However, the resulting field strength is the same as the one obtained above, as shown in the Appendix.

## 5 Discussion

The twistorial formalism we have presented here firmly establishes the geometric foundations of the Newman-Penrose map. Implicit in this construction is the fact that any SNGC on

Minkowski space can be given a purely twistorial definition in terms of the vanishing set of a homogeneous and holomorphic function defined on twistor space [42, 43]. In particular, for any Kerr-Schild spacetime with an expanding SNGC, we have defined a coordinate-independent construction for both the operator $\hat{\kappa}$ and the complex scalar $\Psi$ in terms of twistor variables.

These quantities are then used to define the self-dual gauge field in the Newman-Penrose map, $A = \hat{\kappa}\Psi$, which we showed in previous work [16] coincides with other double copy prescriptions. The close relationship between twistor theory and the geometry of SNGCs is at the heart of this construction, and given the distinguished role that shear-free rays play in several of the approaches to the classical double copy, it is no wonder that twistors are increasingly being recognized as a useful tool in this setting. The work we presented here was partially motivated by applications of twistor theory to the Weyl double copy [46, 47], where the Penrose transform was of central importance; however our investigations seem to run somewhat orthogonal to those references, and it would be very desirable to understand more clearly the relationship between the two approaches.

Given the work that has been done on extending the classical double copy on maximally symmetric backgrounds [20, 21], it could also be interesting to consider maximally symmetric extensions of the twistorial formulation of the Newman-Penrose map. Since (A)dS shares the same twistor space as flat spacetime [59], and since the property of being an SNGC is conformally invariant, it seems reasonable to expect that much of the formalism we have developed here can be adapted to the maximally symmetric case. It would also be interesting to extend this formalism to arbitrary asymptotically flat spacetimes using the generalized Kerr Theorem of [60].

# Acknowledgements

We thank Gilly Elor, Sylvia Nagy, and Paul Tod for discussions and Jake Herczeg for his help designing figure 1. KF acknowledges support from Simons Foundation Award Number 658908. The work of MG is supported by the LDRD program at Los Alamos National Laboratory and by the U.S. Department of Energy, Office of High Energy Physics, under Contract No. DE-AC52-06NA25396.

# A  SNGC Normalization

The point of this Appendix is to show that no loss of generality occurs in normalizing the coefficient of $du$ in the expression (29) for $l_{AA'}dx^{AA'}$ to be equal to one. To show this, we rescale $l_{AA'}dx^{AA'}$ so that its coefficient of $dv$ is normalized to one. Next, we show that the twistorial Newman-Penrose map defined in terms of this rescaled vector gives rise to the same field-strength as in (12).

After rescaling, we have

$$l_{AA'}dx^{AA'} = dv + \Phi^{-1}d\zeta + \bar{\Phi}^{-1}d\bar{\zeta} + (\Phi\bar{\Phi})^{-1}du, \tag{45}$$

where $\Phi$ satisfies (7). Then for a null twistor $Z = (\omega^A, \pi_{A'})$, the normalization condition

$$l_{AA'}\bar{\pi}^A\pi^{A'} = 1 \tag{46}$$

implies $\bar{\pi}^A\pi^{A'}\partial_{AA'} = \partial_v$. Expanding $\omega^A$ and $\pi_{A'}$ as in (31), the condition (46) leads to

$$|\gamma|^2 = 1, \qquad \delta = 0. \tag{47}$$

Since $Z$ is null, $\omega^A \bar{\pi}_A = ia$ implies $\beta = i\gamma a$, giving

$$\omega^A = \alpha o^A + i\gamma a \iota^A, \qquad \pi_{A'} = \gamma \bar{o}_{A'}. \tag{48}$$

The $\alpha$-plane associated to $Z$ has the same form as (35), and here is defined by $u = u_0$ and $\zeta = \zeta_0$, where we have identified $a = u_0$, $\alpha = i\gamma\zeta_0$. This leads to the tangent bivector

$$\tau = \frac{1}{\omega^A \bar{\pi}_A} p \wedge q = \gamma^2 \partial_\nu \wedge \partial_{\bar{\zeta}}, \tag{49}$$

which in turn leads to

$$\hat{\kappa} = \frac{Q}{2\pi\epsilon_0} \gamma^2 (d\zeta \partial_\nu + du \partial_{\bar{\zeta}}). \tag{50}$$

Once expanded in a coordinate spinor basis, the operator $\hat{\kappa}$ has a different expression than the one in (9). One might then expect the resulting field strength $F$ to be different too. However, the associated scalar $\Psi$ is also modified, and these two effects compensate such that the field strength is unchanged.

The definition of $\Psi$ (43) in terms of the twistor (48) is

$$\Psi = -\bar{\gamma}^2 \Phi^{-1} + \frac{\bar{\alpha}\bar{\gamma}}{ia}, \tag{51}$$

which gives for the field strength associated with the gauge field $A = \hat{\kappa}\Psi$,

$$F = -\frac{Q}{2\pi\epsilon_0} \left( \Phi_{,u\zeta}(du \wedge dv + d\zeta \wedge d\bar{\zeta}) + \Phi_{,uu} du \wedge d\bar{\zeta} - \Phi_{\zeta\zeta} dv \wedge d\zeta \right), \tag{52}$$

which is identical to (12).

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
