# Peer review of "Twistor Space Origins of the Newman-Penrose Map"

_SciPost Physics Core, doi:SciPost Phys. 13, 099 (2022)_

## Round 2 · Referee Report · Anonymous (Referee 6) · 2022-4-19

Report
The paper explores the geometrical description in twistor space of the Newman-Penrose map, which is closely related to the classical double copy. The authors discover an interesting geometrical origin of the spin-raising operator $\hat{k}$ which appears in the classical self-dual double copy and in the Newman-Penrose map. While the paper is well-written, some important points should be addressed before I can recommend for publication:
-Under Eq. 2.7 it is mentioned given $\Phi$ specified by 2.6, the solution to Einstein’s equation is completely specified except for its total mass. There are several definitions for a mass in General Relativity, which one is this referring to?
-The building blocks of the Newman-Penrose map are written in terms of spinors, some that naturally arise as components of a null twistor ($\omega^A,\pi_{A’}$) and others that don’t ($l_{AA’}$). Meanwhile, it is mentioned in the abstract, introduction, and discussion that this alternative definition is given in terms of quantities defined on both spacetime and twistor space. The results in Eqs. 4.15 and 4.16 don’t support this claim. They are not written in terms of twistors and include $l_{AA’}$ which is not defined in twistor space. The claim must be clearly proven under these equations or it should be rephrased.
-In Eq. 4.17 an alternative definition of the gauge field in the Newman-Penrose map is considered. It is mentioned that it has no dependence on the phase $\delta$ and that it is equivalent to the definition in 2.9. This is not true unless you ignore the extra constant term. While that term does not contribute to the field strength, it is still present in the gauge field. To make those claims one would have to do a gauge transformation to remove the extra term. These shortcomings should be clarified instead of swept under the carpet.
This paper could be suitable for publication in SciPost Physics Core after the concerns mentioned above have been addressed.

---

## Round 2 · Referee Report · Anonymous (Referee 5) · 2022-6-2

Report
This paper concerns the application of twistor methods to the study of the “double copy” relating quantities in gauge and gravity theories. The authors reconsider a previously presented proposal for generating self-dual single copies known as the Newman-Penrose map. Gauge fields in this picture are comprised of a certain differential operator acting on a given scalar function. The authors show that both of these ingredients can be given geometric definitions motivated by twistor theory, such that ambiguities in the definitions cancel out in the gauge field. Given the use of twistor methods in other research works concerning the double copy, the present paper is a highly useful addition to the literature, and I recommend publication.
I note that the first referee has a number of comments, which I agree need addressing. But I have a remark that the authors may also wish to comment on. They wonder how their analysis is related to other twistor-related work on the double copy, which relies on the well-known Penrose transform from twistor space to position space. I suspect that the answer lies in their use of the Kerr theorem, which states that shear-free null geodesic congruences (SNGCs) such as those entering Kerr-Schild geometries can be defined by the vanishing of a function of twistor variables. In the Penrose transform approach to the double copy, the contour integral in the Penrose transform picks out the residue of poles in the integrand, which in turn enforces the vanishing of certain twistor functions. So the various approaches seem conceptually related, at least in principle. It would be interesting to know if the authors agree or not with this suggestion, but they should not feel obliged to comment.

---

## Round 2 · Referee Report · Anonymous (Referee 4) · 2022-6-15

Strengths
1.) A novel approach to classical double copy making use of classic results in twistor theory (including the Kerr theorem and spin-raising operators).
2.) Clearly written.
Weaknesses
1.) This shares the main weakness of most classical double copy approaches, insofar as it relies on algebraic speciality on the gravitational side (and therefore cannot teach us something new about the space of solutions in GR).
Report
This paper builds on prior work by the authors in defining a 'Newman-Penrose map' which takes Kerr-Schild solutions in general relativity defined using a shear-free null geodesic vector, and maps them to self-dual solutions of the Maxwell equations. In particular, the authors provide a twistor interpretation for this map, demonstrating that various ingredients in the prescription arise naturally in the twistor description of shear-free null geodesic congruences. I think that this is an interesting and novel paper, and deserves to be published in SciPost Physics Core; although prior to publication, the authors should address the issues raised by the other referees.
I also have a (possibly naive) question for the authors, which is as much about their prior work on this topic as it is about this particular paper. Here, the Maxwell field is defined by $A=\hat{k}\Phi$, which results in a complex (in Lorentzian signature), self-dual abelian gauge field. However, a real-valued gauge field can also be obtained by taking $A=\frac{1}{2}\hat{k}\Phi+\frac{1}{2}\hat{\bar{k}}\bar{\Phi}$, and the resulting field strength is a non-chiral solution to the Maxwell equations. (Of course, this isn't true for non-abelian gauge fields, where the SD/ASD parts of the potential will mix in commutators for the field strength, but for Maxwell it is fine.) Furthermore, it seems that this non-chiral Maxwell field coincides with that produced by the 'Kerr-Schild' version of classical double copy. For instance, in the case of Schwarzschild, this will lead to a gauge potential which is real and gauge-equivalent to Coulomb. I haven't checked other examples (e.g., Kerr), but expect that something similar will happen there.
This suggests that the Newman-Penrose map, as stated, is just the chiral half of another map which is really equivalent to the Kerr-Schild double copy, but refined by the shear-free condition. Have the authors considered this, and if so, is there some reason why this is less interesting than their chiral version of the Newman-Penrose map?
Requested changes
1.) The points raised by the other referees, particularly referee 1, should be addressed.
2.) On page 1, the authors say that there is a growing body of evidence that double copy can be "...extended to all orders in perturbation theory [4-11]." This may be nitpicking, but I think that this is an over-optimistic statement: in terms of what is actually known, I think it is fair to say that double copy can be extended up to 5-loops, for the 4-point amplitude. A more accurate statement might be "...extended beyond tree-level in perturbation theory [4-11]."
3.) For completeness, the authors should define the $Q$ and $\varepsilon_0$ appearing in equation (2.8).

---

## Round 3 · Referee Report · Anonymous (Referee 2) · 2022-7-8

Report

I am satisfied that the authors have dealt with my previous comments, and recommend publication.

---

## Round 3 · Referee Report · Anonymous (Referee 3) · 2022-7-12

Report

The authors have addressed all of the points raised in my previous report (as well as those of the other reviewers). I now think that this paper is ready to be published in SciPost Physics Core.

---

## Round 3 · Referee Report · Anonymous (Referee 1) · 2022-7-13

Report

The author's have now addressed appropriately all the comments and concerns and the manuscript is now suitable for publication.

---

## Round 3 · Author Response

Dear editor and referees,

We thank the referees for their time and very useful feedback. We have incorporated the
suggestions provided and hope the current version of the paper can be published. Below we
have enumerated the points brought up by the referees as well as our responses to them.

Sincerely,

Kara Farnsworth, Michael Graesser and Gabriel Herczeg

---

## Round 3 · List of Changes

Referee 1:

  • Comment 1: ``Under Eq. 2.7 it is mentioned given $\Phi$ specified by 2.6, the solution to Einstein’s equation is completely specified except for its total mass. There are several definitions for a mass in General Relativity, which one is this referring to?"

We thank the referee for this comment and apologize for our sloppy language. The referee is correct that there are several definitions of mass'' and the notion itself is subtle and has a long history. Bymass,'' here we mean a constant integration parameter that appears in solving Einstein's equations, and not necessarily to the ADM, Bondi or Komar masses, for instance. To avoid confusion, we have modified the text to:

``When the spacetime solves the vacuum Einstein equations, the function $V$ can be solved for in terms of $\Phi$, so that any $\Phi$ satisfying (2.6) completely specifies the solution up to constants of integration."

  • Comment 2: ``The building blocks of the Newman-Penrose map are written in terms of spinors, some that naturally arise as components of a null twistor $(\omega^A, \pi_{A'})$ and others that don’t $(l_{AA'})$. Meanwhile, it is mentioned in the abstract, introduction, and discussion that this alternative definition is given in terms of quantities defined on both spacetime and twistor space. The results in Eqs. 4.15 and 4.16 don’t support this claim. They are not written in terms of twistors and include $l_{AA'}$ which is not defined in twistor space. The claim must be clearly proven under these equations or it should be rephrased."

We thank the referee for pointing out this lack of precision in our word choice. It was not our intent to suggest that all quantities in the definition of the Newman-Penrose map are defined on both Minkowski and twistor space, but rather that the quantities involved are all defined on one of the two spaces. That being said, shear-free null geodesic congruences on Minkowski space do have a very natural connection with twistor space furnished by the Kerr theorem. We have attempted to clarify this point in the abstract, where we have made the following changes:

``Here, we give an alternative definition of this correspondence in terms of quantities that are defined naturally on twistor space, and a shear-free null geodesic congruence on Minkowski space whose twistorial character is articulated by the Kerr theorem."

Of course, as we discussed in section (3), a null twistor corresponds to a real null geodesic on Minkowski space. We have also added a figure in section 3.2 to help illustrate this relationship.

And in the concluding discussion section we have added the following sentence that reiterates the twistorial origins of the shear-free null congruence on Minkowski space, which is used implicitly in our construction, as well as references to the literature discussing this idea:

``Implicit in this construction is the fact that any SNGC on Minkowski space can be given a purely twistorial definition in terms of the vanishing set of a homogeneous and holomorphic function defined on twistor space [42,43]."

  • Comment 3: ``In Eq. 4.17 an alternative definition of the gauge field in the Newman-Penrose map is considered. It is mentioned that it has no dependence on the phase $\delta$ and that it is equivalent to the definition in 2.9. This is not true unless you ignore the extra constant term. While that term does not contribute to the field strength, it is still present in the gauge field. To make those claims one would have to do a gauge transformation to remove the extra term. These shortcomings should be clarified instead of swept under the carpet."

We appreciate the detailed feedback of the referee, however we want to emphasize that while the scalar functions $\Psi$ and $\Phi$ do indeed differ by an additional constant and a phase, the gauge fields $A = \hat{\kappa}\Psi$ (4.17) and $A = \hat{k}\Phi$ (2.9) do not. To clarify this point, we have distinguished the original, coordinate-dependent definition of the spin-raising operator $\hat{k}$ from the twistorial version, which we now denote as $\hat{\kappa}$ in section 4, and we have emphasized below eq (4.16) that $\frac{\bar{\delta}\bar{\beta}}{ia}$ is a constant. In particular, we have made the following changes to the text:

`` The first term in $\Psi$ differs from $\Phi$ by the constant phase $\bar{\delta}^2$, complementary to the phase of $\hat{\kappa}$. The second term is a constant, so it is annihilated by $\hat{\kappa}$.

Referee 2:

  • Comment 1: ``I note that the first referee has a number of comments, which I agree need addressing. But I have a remark that the authors may also wish to comment on. They wonder how their analysis is related to other twistor-related work on the double copy, which relies on the well-known Penrose transform from twistor space to position space. I suspect that the answer lies in their use of the Kerr theorem, which states that shear-free null geodesic congruences (SNGCs) such as those entering Kerr-Schild geometries can be defined by the vanishing of a function of twistor variables. In the Penrose transform approach to the double copy, the contour integral in the Penrose transform picks out the residue of poles in the integrand, which in turn enforces the vanishing of certain twistor functions. So the various approaches seem conceptually related, at least in principle. It would be interesting to know if the authors agree or not with this suggestion, but they should not feel obliged to comment."

For the first comment, please see the responses to Referee 1.

We thank the Referee for their second set of observations. We agree with the Referee that the Newman-Penrose map is conceptually related to the use of the Penrose transform in other literature. In previous unpublished work we noticed that, at least for the Schwarzchild solution, the two possible choices for the harmonic function $\Phi$ in the Newman-Penrose map, corresponding to the two choices of incoming or outgoing SNGCs, happen to coincide with the poles of the twistor function, as well to the components of the principal spinors in the factorization of the Weyl tensor; specifically Eqns. (77) and (80) of ``The Weyl Double Copy from Twistor Space'' [arXiv:2103.16441]. We are currently trying to understand our observation better, and whether the same pattern generalizes to the Kerr and other solutions.

Referee 3:

  • Comment 1: ``I also have a (possibly naive) question for the authors, which is as much about their prior work on this topic as it is about this particular paper. Here, the Maxwell field is defined by $A = \hat{k}\Phi$, which results in a complex (in Lorentzian signature), self-dual abelian gauge field. However, a real-valued gauge field can also be obtained by taking $A =\frac{1}{2} \hat{k}\Phi+\frac{1}{2} \hat{\bar{k}}\bar{\Phi}$, and the resulting field strength is a non-chiral solution to the Maxwell equations. (Of course, this isn't true for non-abelian gauge fields, where the SD/ASD parts of the potential will mix in commutators for the field strength, but for Maxwell it is fine.) Furthermore, it seems that this non-chiral Maxwell field coincides with that produced by the 'Kerr-Schild' version of classical double copy. For instance, in the case of Schwarzschild, this will lead to a gauge potential which is real and gauge-equivalent to Coulomb. I haven't checked other examples (e.g., Kerr), but expect that something similar will happen there.

This suggests that the Newman-Penrose map, as stated, is just the chiral half of another map which is really equivalent to the Kerr-Schild double copy, but refined by the shear-free condition. Have the authors considered this, and if so, is there some reason why this is less interesting than their chiral version of the Newman-Penrose map?"

The referee is correct that the Maxwell field defined by the Newman-Penrose map results in a complex, self-dual gauge field. However when we compare our results to other versions of the classical double copy, we take the real part of our gauge field, which is exactly what the referee suggests, i.e. $A =\frac{1}{2} \hat{k}\Phi+\frac{1}{2} \hat{\bar{k}}\bar{\Phi}$. It is this real gauge field that exactly matches (up to gauge transformations and some subtleties explained in the paper) the Kerr-Schild double copy examples given in our first paper: Schwarzschild, Kerr and the Photon Rocket. We hope to understand in detail the precise relationship between our map and the Kerr-Schild double copy in future work, and hope the twistor formulation presented in the current paper can facilitate this. For more on this last statement, please see the second response to Referee 2.

  • Comment 2: ``The points raised by the other referees, particularly referee 1, should be addressed."

Please see responses to Referee 1.

  • Comment 3: ``On page 1, the authors say that there is a growing body of evidence that double copy can be "...extended to all orders in perturbation theory [4-11]." This may be nitpicking, but I think that this is an over-optimistic statement: in terms of what is actually known, I think it is fair to say that double copy can be extended up to 5-loops, for the 4-point amplitude. A more accurate statement might be "...extended beyond tree-level in perturbation theory [4-11].""

We thank the referee for this clarification and have implemented this suggestion.

  • Comment 4: ``For completeness, the authors should define the $Q$ and $\epsilon_0$ appearing in equation (2.8)."

We thank the referee for this clarification and have implemented this suggestion by adding the following sentence after Eq. (2.8):

``Here we have included the constants $Q$, representing the charge, and $\epsilon_0$, representing the vacuum permittivity, to directly map to solutions of Maxwell’s equations."

---

## Editorial Decision

published